# Genetic risk, adherence to healthy lifestyle and acute cardiovascular and thromboembolic complications following SARS-COV-2 infection

Junqing Xie [1,10], Yuliang Feng[2,3,10], Danielle Newby[1], Bang Zheng[4], Qi Feng[5], Albert Prats-Uribe [1], Chunxiao Li[6], Nicholas J. Wareham [6], R. Paredes[7,8] & Daniel Prieto-Alhambra [1,9] ✉

Current understanding of determinants for COVID-19-related cardiovascular and thromboembolic (CVE) complications primarily covers clinical aspects with limited knowledge on genetics and lifestyles. Here, we analysed a prospective cohort of 106,005 participants from UK Biobank with confirmed SARS-CoV-2 infection. We show that higher polygenic risk scores, indicating individual's hereditary risk, were linearly associated with increased risks of post-COVID-19 atrial fibrillation (adjusted HR 1.52 [95% CI 1.44 to 1.60] per standard deviation increase), coronary artery disease (1.57 [1.46 to 1.69]), venous thromboembolism (1.33 [1.18 to 1.50]), and ischaemic stroke (1.27 [1.05 to 1.55]). These genetic associations are robust across genders, key clinical subgroups, and during Omicron waves. However, a prior composite healthier lifestyle was consistently associated with a reduction in all outcomes. Our findings highlight that host genetics and lifestyle independently affect the occurrence of CVE complications in the acute infection phrase, which can guide tailored management of COVID-19 patients and inform population lifestyle interventions to offset the elevated cardiovascular burden post-pandemic.

Cardiovascular disease is the leading cause of death globally[1]. Recently, cardiovascular mortality and morbidity have risen further due to the direct and indirect consequences of the COVID-19 pandemic[2,3]. It is expected that the repercussions and long-term sequelae of COVID-19 could further increase the cardiovascular burden to an unprecedented level[4].

At the individual level, preventing life-threatening cardiovascular and thromboembolic complications (CVE) is crucial during the treatment of patients with COVID-19. However, a clinical challenge remains to accurately identify individuals at risk to warrant intensive surveillance or targeted pharmacological prophylaxis. For instance, although prophylactic anticoagulation has been recommended for hospitalised

[1]Centre for Statistics in Medicine and NIHR Biomedical Research Centre Oxford, NDORMS, University of Oxford, Oxford, UK. [2]Botnar Research Centre, Nuffield Department of Orthopaedics, Rheumatology and Musculoskeletal Sciences, University of Oxford, Oxford, UK. [3]Department of Pharmacology, School of Medicine, Southern University of Science and Technology, Shenzhen, Guangdong, China. [4]Department Non-communicable Disease Epidemiology, London School of Hygiene & Tropical Medicine, London, UK. [5]Nuffield Department of Population Health, University of Oxford, Oxford, UK. [6]Medical Research Council Epidemiology Unit, University of Cambridge, Cambridge, UK. [7]Department of Infectious Diseases Department & irsiCaixa AIDS Research Institute, Hospital Universitari Germans 13 Trias i Pujol, Catalonia, Spain. [8]Center for Global Health and Diseases, Department of Pathology, Case Western Reserve University School of Medicine, Cleveland, OH, US. [9]Department of Medical Informatics, Erasmus Medical Center University, Rotterdam, Netherlands. [10]These authors contributed equally: Junqing Xie, Yuliang Feng. ✉e-mail: Daniel.prietoalhambra@ndorms.ox.ac.uk

patients with COVID-19[5], evidence for its use is vastly conflicting for more critical ICU patients and milder ambulatory patients with COVID-19[6-8].

Patient's features, including age, sex, and obesity, are recognised general risk factors for severe COVID-19, such as hospital admission and mechanical ventilation. Although they are helpful in informing clinical practice, they are not specific to CVE complications following the infection. In contrast, polygenic risk scores (PRS), a sum of genetic risk for a particular trait, have been recently proposed as a promising tool for precision medicine and early risk stratification[9-11]. It is not yet known whether the genetic susceptibility to chronic cardiovascular diseases, as measured by the PRS, can also predispose the occurrence of clinically relevant CVE complications during the acute phase of COVID-19.

In addition, effective public health interventions are urgently needed to reduce the population's cardiovascular burden, particularly in light of surging COVID-19 infections after the removal of most early restrictions (e.g., lockdown and social distance). The US Preventive Service Task Force updated its recommendations in 2022, promoting healthy behaviour counselling for all adults as a national strategy for primary cardiovascular prevention[12]. However, all current clinical and public health guidelines[13,14] lack insights into the potential role of healthy lifestyle modifications in alleviating COVID-19 cardiovascular complications, likely due to a paucity of evidence.

We aimed to assess the association between host genetics, lifestyle factors, and their combined effects on the risk of four major CVE events within 90 days after COVID-19 diagnosis.

## Results

### Population characteristics

Table 1 shows the baseline characteristics of the whole UK Biobank participants eligible for this study (n = 407,453) and of all participants with COVID-19 during the study period from 01/03/2020 to 30/09/2022 (n = 106,005), overall and stratified by their genetic risk of AF. The average (SD) age of the COVID-19 cohort was 67.68 (8.26) years. Among them, 54.8% were female and the majority were White ethnicity (89.4%). The prevalence of the nine prespecified unhealthy lifestyle factors ranged from 8.7% for the smoking status and 53.0% for the red meat intake. In total, 7.4%, 38.0% and 54.7% of infected individuals lived an unfavourable, moderate and favourable composite lifestyle, respectively. These proportions were highly consistent with those of the source UK Biobank population. All study covariates, except ethnicity, were distributed similarly across different PRS strata, illustrating independence between the genetic and lifestyle factors. The baseline characteristics by the genetic risk for CAD, VTE and ISS were similar to those by the AF genetic score (Supplementary Tables 1–3).

After COVID-19 infection, 1397 AF, 733 CAD, 244 VTE, and 104 ISS events occurred during the 90-day follow-up period. It accounted for 1.31%, 0.69%, 0.23%, and 0.09% of the population, respectively, with incidence rates (IR) corresponding to 56.05 per 1,000 person-years for AF, 29.30 for CAD, 9.73 for VTE, and 4.14 for ISS (Supplementary Table 4).

### Genetic risk and acute COVID-19 CVE complications

Figure 1 of Kaplan−Meier cumulative incidence curves shows that most of CVE complications (AF, CAD, and VTE) increased dramatically within the first 30 days after infection, and flatten gradually later. Early separation of risk was consistently observed across the three genetic risk subgroups and continued to diverge over time. Compared to participants with a lower genetic risk, those with a higher genetic risk had elevated incidences of acute CVE events in a dose-dependent manner. Specifically, the adjusted HR of intermediate vs low genetic risk was 1.95 (95% 1.62 to 2.34) for AF, 1.63 (1.28 to 2.08) for CAD, 1.35 (0.93 to 1.96) for VTE and 1.88 (0.99 to 3.58) for ISS complications, whereas was 3.45 (2.85 to 4.18), 3.33 (2.59 to 4.28), 2.12 (1.41 to 3.18) and

2.66 (1.33 to 5.32), correspondingly, in comparison with high genetic risk (Table 2). These categorical genetic risks remained consistent with infections during both the pre-Omicron and Omicron periods.

A normal distribution was observed in all four PRSs (Supplementary Fig. 1). Higher genetic risk in the form of continuous PRS was found to be associated with an increased acute risk of COVID-19 CVE complication, with the HR per standard deviation increase equal to 1.52 (1.44 to 1.60) for AF, 1.57 (1.46 to 1.69) for CAD, and 1.33 (1.18 to 1.50) for VTE and 1.27 (1.05 to 1.55) for ISS (Table 2). These associations were likely to be linear, with all P-values for the linear term <0.001 and for the non-linear term >0.05 (Supplementary Fig. 1). Importantly, similar PRS associations were seen in individuals infected with either the pre-Omicron or Omicron variants (Table 2).

The positive association between each PRS and its relevant COVID-19 complication held across the different subgroups of clinical interest, despite variation in baseline risks and effect sizes. (Fig. 2). For example, the IR of AF was 85.25 per 1000 person-years among participants with COVID-19 aged 65 years or older, more than 8-fold higher than those younger than 65 years (IR: 10.84 per 1000 person-years), but the HR was almost the same for both groups. Also, it is notable that the association of PRS was more pronounced among recent users of antithrombotics than non-users(1.92 vs 1.36 for AF, P-interaction < 0.01 and 2.18 vs 1.29 for CAD, P-interaction < 0.01), although they, as expected, had much higher background risks (IR: 241.01 for AF and 171.47 for CAD per 1,000 person-years) compared with non-users (IR: 29.74 and 8.96 respectively). Importantly, the genetic risk persisted for all four CVE complications among fully vaccinated individuals experiencing breakthrough COVID-19 infection.

### Healthy lifestyle and acute COVID-19 CVE complications

Compared to those with a less healthy lifestyle, COVID-19 patients who adhered to healthier habits prior to infection had a significantly lower risk of CVE complications, with a HR of continuous HLS for AF corresponding to 0.89 (95% CI 0.86 to 0.93), for CAD of 0.87 (95% CI 0.82 to 0.91), for VTE of 0.88 (95% CI 0.80 to 0.96), and for ISS of 0.86 (95% CI 0.75 to 0.99) within 90 days following the infection(Table 2). The Kaplan-Meier curves in Fig. 1 show that patients with a favourable lifestyle had reduced cumulative incidences compared to those with a moderate or unfavourable lifestyle, resulting in an adjusted HR of 0.77 (0.64 to 0.92) and 0.66 (0.56 to 0.79) for AF, 0.71 (0.57 to 0.89) and 0.57 (0.45 to 0.71) for CAD, 0.86 (0.56 to 1.31) and 0.65 (0.42 to 1.00) for VTE, and 0.56 (0.31 to 1.02) and 0.46 (0.25 to 0.83) for ISS. The protective associations of composite healthy lifestyle with acute COVID-19 complications appeared to be attenuated during the Omicron than pre-Omicron periods. For each of the nine individual healthy lifestyle habits, all were associated with either lower or non-differential CVE risks after COVID-19, except for the alcohol drinking factor: "≤ 4 times week" vs "Daily or almost daily" that was marginally associated with a higher CAD risk, possibly due to a type 1 (false positive) error under the multiple statistical testing (9 exposures * 4 outcomes = 28 comparisons) (Supplementary Fig. 2).

### Joint associations between genetic and lifestyle factors

A considerable additive association was observed between genetic and lifestyle factors in relation to AF and CAD complications following COVID-19 (Fig. 3). The adjusted cumulative incidence was highest among individuals possessing high genetic risk and an unfavourable lifestyle, while it was lowest for those with low genetic risk and a favourable lifestyle. The adjusted HR among the nine subgroups varied from 0.73 (95% CI 0.55 to 0.99) to 0.18 (0.13 to 0.26) for AF and 0.92 (0.62 to 1.36) to 0.20 (0.13 to 0.33) for CAD. Although the point estimates appeared to be additive also for VTE and ISS outcomes, less robust statistical significance of a joint association was found.

**Table 1 | Characteristics of participants with COVID-19 overall and in subgroups by the genetic risk of atrial fibrillation**

| Participant characteristics | All eligible participants[a] | Participants with COVID-19 | | | | |
|---|---|---|---|---|---|---|
| | | All | Low genetic risk | Intermediate genetic risk | High genetic risk | SMD |
| Number | 407453 | 106005 | 21201 | 63603 | 21201 | |
| Mean age, year (SD) | 69.46 (8.14) | 67.68 (8.26) | 67.61 (8.22) | 67.72 (8.27) | 67.64 (8.25) | 0.009 |
| Sex, No. (%) | | | | | | |
| Female | 225700 (55.4) | 58045 (54.8) | 11663 (55.0) | 34822 (54.7) | 11560 (54.5) | 0.007 |
| Male | 181752 (44.6) | 47960 (45.2) | 9538 (45.0) | 28781 (45.3) | 9641 (45.5) | |
| Ethnicity, No. (%) | | | | | | |
| White | 375824 (92.2) | 94821 (89.4) | 19023 (89.7) | 57214 (90.0) | 18584 (87.7) | 0.049 |
| Other ethnic groups | 31629 (7.8) | 11184 (10.6) | 2178 (10.3) | 6389 (10.0) | 2617 (12.3) | |
| Obesity, No. (%) | | | | | | |
| BMI < 30 | 310852 (76.3) | 80339 (75.8) | 15764 (74.4) | 48305 (75.9) | 16270 (76.7) | 0.037 |
| BMI ≥ 30 | 96601 (23.7) | 25666 (24.2) | 5437 (25.6) | 15298 (24.1) | 4931 (23.3) | |
| Socioeconomic deprivation[b], mean (SD) | 17.47 (13.75) | 17.13 (13.19) | 17.18 (13.21) | 17.14 (13.20) | 17.04 (13.15) | 0.007 |
| Education categories[c], No. (%) | | | | | | |
| I | 72698 (17.8) | 13808 (13.0) | 2813 (13.3) | 8345 (13.1) | 2650 (12.5) | 0.033 |
| II | 111099 (27.3) | 30324 (28.6) | 6207 (29.3) | 18201 (28.6) | 5916 (27.9) | |
| III | 45501 (11.2) | 12762 (12.0) | 2545 (12.0) | 7709 (12.1) | 2508 (11.8) | |
| IV | 20685 (5.1) | 5113 (4.8) | 1016 (4.8) | 3050 (4.8) | 1047 (4.9) | |
| V | 157469 (38.6) | 43998 (41.5) | 8620 (40.7) | 26298 (41.3) | 9080 (42.8) | |
| Individual lifestyle factors, No. (%) | | | | | | |
| Smoking status | | | | | | |
| Never or past | 367963 (90.3) | 96827 (91.3) | 19364 (91.3) | 58069 (91.3) | 19394 (91.5) | 0.004 |
| Current | 39490 (9.7) | 9178 (8.7) | 1837 (8.7) | 5534 (8.7) | 1807 (8.5) | |
| Alcohol intake | | | | | | |
| ≤ 4 times week | 324878 (79.7) | 85137 (80.3) | 17028 (80.3) | 51040 (80.2) | 17069 (80.5) | 0.004 |
| Daily or almost daily | 82575 (20.3) | 20868 (19.7) | 4173 (19.7) | 12563 (19.8) | 4132 (19.5) | |
| Physical activity | | | | | | |
| ≥150 min per week of MIPA or ≥75 min per week of VIPA | 344442 (84.5) | 88608 (83.6) | 17694 (83.5) | 53186 (83.6) | 17728 (83.6) | 0.003 |
| <150 min per week of MIPA & <75 min per week of VIPA | 63011 (15.5) | 17397 (16.4) | 3507 (16.5) | 10417 (16.4) | 3473 (16.4) | |
| Television viewing time | | | | | | |
| <4 h/day | 293745 (72.1) | 79656 (75.1) | 15801 (74.5) | 47761 (75.1) | 16094 (75.9) | 0.021 |
| ≥ 4 h/day | 113708 (27.9) | 26349 (24.9) | 5400 (25.5) | 15842 (24.9) | 5107 (24.1) | |
| Sleep duration | | | | | | |
| ≥7 & ≤9 h/day | 301120 (73.9) | 79162 (74.7) | 15842 (74.7) | 47402 (74.5) | 15918 (75.1) | 0.008 |
| <7 or >9 h/day | 106333 (26.1) | 26843 (25.3) | 5359 (25.3) | 16201 (25.5) | 5283 (24.9) | |
| Fruit and vegetable intake | | | | | | |
| ≥ 400 g/day | 326692 (80.2) | 84622 (79.8) | 16978 (80.1) | 50691 (79.7) | 16953 (80.0) | 0.006 |
| <400 g/ day | 80761 (19.8) | 21383 (20.2) | 4223 (19.9) | 12912 (20.3) | 4248 (20.0) | |
| Oily fish intake | | | | | | |
| ≥1 portion/week | 229373 (56.3) | 56737 (53.5) | 11289 (53.2) | 34032 (53.5) | 11416 (53.8) | 0.008 |
| <1 portion/week | 178080 (43.7) | 49268 (46.5) | 9912 (46.8) | 29571 (46.5) | 9785 (46.2) | |
| Red meat intake | | | | | | |
| ≤3 portion/week | 187802 (46.1) | 49840 (47.0) | 10131 (47.8) | 29818 (46.9) | 9891 (46.7) | 0.015 |
| >3 portion/week | 219651 (53.9) | 56165 (53.0) | 11070 (52.2) | 33785 (53.1) | 11310 (53.3) | |
| Processed meat intake | | | | | | |
| ≤1 portion/week | 281938 (69.2) | 72015 (67.9) | 14370 (67.8) | 43224 (68.0) | 14421 (68.0) | 0.003 |
| >1 portion/week | 125515 (30.8) | 33990 (32.1) | 6831 (32.2) | 20379 (32.0) | 6780 (32.0) | |
| Composite healthy lifestyle, No. (%) | | | | | | |
| Unfavourable | 30746 (7.5) | 7816 (7.4) | 1584 (7.5) | 4696 (7.4) | 1536 (7.2) | 0.013 |
| Moderate | 154864 (38.0) | 40238 (38.0) | 8020 (37.8) | 24278 (38.2) | 7940 (37.5) | |
| Favourable | 221843 (54.4) | 57951 (54.7) | 11597 (54.7) | 34629 (54.4) | 11725 (55.3) | |

*BMI* body mass index, *MIPA* moderate intensity physical activity, *VIPA* vigorous intensity physical activity, *SMD* standardised mean difference.
[a]All eligible participants in UK Biobank included those who survived when this study began (March 1, 2020).
[b]High score indicates higher levels of deprivation.
[c]Education category I includes self-reported "None of the above" and "Prefer not to answer", II includes "CSEs or equivalent" and "O levels/GCSEs or equivalent", III includes "A levels/AS levels or equivalent", IV includes "Other professional qualifications e.g.: nursing, teaching", and V includes "NVQ or HND or HNC or equivalent" and "College or University degree."

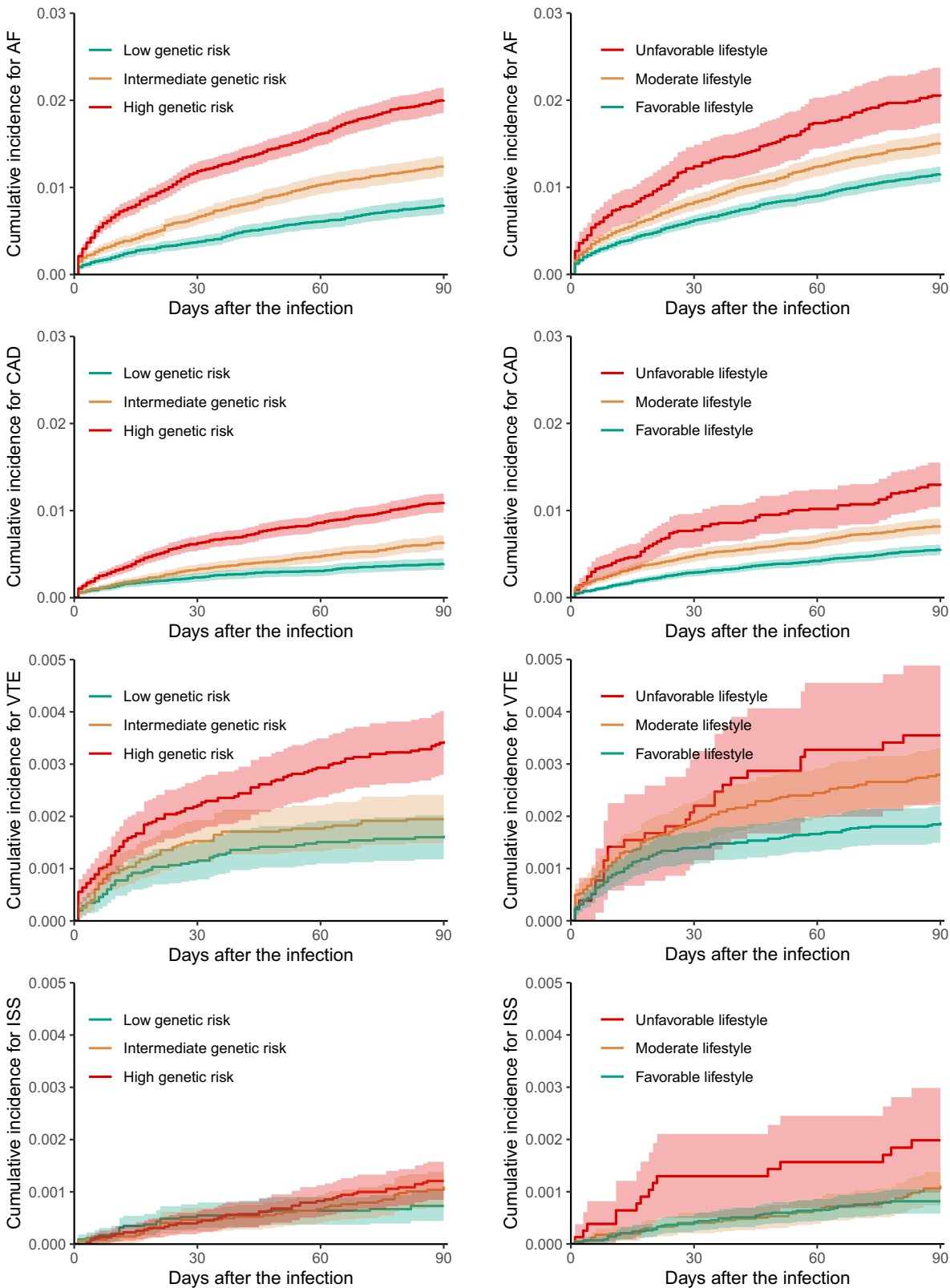

**Fig. 1 | Unadjusted cumulative incidence of post-COVID-19 cardiovascular and thromboembolic complications within 90 days.** AF atrial fibrillation, CAD coronary artery disease, VTE venous thromboembolism, ISS ischaemic stroke. The upper limit of the *y*-axis is different between plots for better visualisation. The shadow of curves represents 95% confidence interval.

**Table 2 | Associations between the genetic risk, healthy lifestyle and post-COVID-19 cardiovascular and thromboembolic complications in patients with any, prior Omicron, and Omicron infection**

| | Genetic risk: Adjusted hazard ratio (95% CI) | | | | Lifestyle: Adjusted hazard ratio (95% CI) | | |
|---|---|---|---|---|---|---|---|
| | Any COVID-19 infection | Prior Omicron infection | Omicron infection | | Any COVID-19 infection | Prior Omicron infection | Omicron infection |
| AF | (case = 1397) | (case = 561) | (case = 836) | AF | (case = 1397) | (case = 561) | (case = 836) |
| Continuous PRS[a] | 1.52 (1.44 to 1.60) | 1.46 (1.35 to 1.59) | 1.55 (1.45 to 1.66) | Continuous HLS[b] | 0.89 (0.86 to 0.93) | 0.86 (0.81 to 0.91) | 0.92 (0.87 to 0.97) |
| Low | 1 [ref] | 1 [ref] | 1 [ref] | Unfavourable | 1 [ref] | 1 [ref] | 1 [ref] |
| Intermediate | 1.95 (1.62 to 2.34) | 1.65 (1.26 to 2.16) | 2.19 (1.71 to 2.80) | Moderate | 0.77 (0.64 to 0.92) | 0.66 (0.51 to 0.85) | 0.88 (0.69 to 1.13) |
| High | 3.45 (2.85 to 4.18) | 2.86 (2.15 to 3.81) | 3.93 (3.04 to 5.08) | Favourable | 0.66 (0.56 to 0.79) | 0.56 (0.43 to 0.72) | 0.78 (0.61 to 1.00) |
| P for trend | <0.001 | <0.001 | <0.001 | P for trend | <0.001 | <0.001 | 0.024 |
| CAD | (case = 733) | (case = 321) | (case = 412) | CAD | (case = 733) | (case = 321) | (case = 412) |
| Continuous PRS[a] | 1.57 (1.46 to 1.69) | 1.59 (1.43 to 1.78) | 1.55 (1.40 to 1.71) | Continuous HLS[b] | 0.87 (0.82 to 0.91) | 0.86 (0.79 to 0.93) | 0.88 (0.82 to 0.94) |
| Low | 1 [ref] | 1 [ref] | 1 [ref] | Unfavourable | 1 [ref] | 1 [ref] | 1 [ref] |
| Intermediate | 1.63 (1.28 to 2.08) | 1.66 (1.14 to 2.42) | 1.61 (1.17 to 2.21) | Moderate | 0.71 (0.57 to 0.89) | 0.67 (0.48 to 0.92) | 0.76 (0.55 to 1.04) |
| High | 3.33 (2.59 to 4.28) | 3.84 (2.61 to 5.64) | 3.02 (2.17 to 4.22) | Favourable | 0.57 (0.45 to 0.71) | 0.54 (0.38 to 0.75) | 0.60 (0.44 to 0.83) |
| P for trend | <0.001 | <0.001 | <0.001 | P for trend | <0.001 | <0.001 | <0.001 |
| VTE | (case = 244) | (case = 166) | (case = 78) | VTE | (case = 244) | (case = 166) | (case = 78) |
| Continuous PRS[a] | 1.33 (1.18 to 1.50) | 1.29 (1.11 to 1.49) | 1.41 (1.15 to 1.74) | Continuous HLS[b] | 0.88 (0.80 to 0.96) | 0.86 (0.77 to 0.96) | 0.94 (0.79 to 1.11) |
| Low | 1 [ref] | 1 [ref] | 1 [ref] | Unfavourable | 1 [ref] | 1 [ref] | 1 [ref] |
| Intermediate | 1.35 (0.93 to 1.96) | 1.31 (0.83 to 2.04) | 1.59 (0.78 to 3.27) | Moderate | 0.86 (0.56 to 1.31) | 0.83 (0.51 to 1.36) | 0.94 (0.41 to 2.13) |
| High | 2.12 (1.41 to 3.18) | 1.88 (1.15 to 3.08) | 2.93 (1.37 to 6.27) | Favourable | 0.65 (0.42 to 1.00) | 0.63 (0.38 to 1.04) | 0.74 (0.32 to 1.68) |
| P for trend | <0.001 | <0.001 | <0.001 | P for trend | 0.021 | 0.057 | 0.316 |
| ISS | (case =104) | (case = 38) | (case = 66) | ISS | (case =104) | (case = 38) | (case = 66) |
| Continuous PRS[a] | 1.27 (1.05 to 1.55) | 1.05 (0.76 to 1.44) | 1.44 (1.12 to 1.84) | Continuous HLS[b] | 0.86 (0.75 to 0.99) | 0.83 (0.66 to 1.05) | 0.89 (0.74 to 1.06) |
| Low | 1 [ref] | 1 [ref] | 1 [ref] | Unfavourable | 1 [ref] | 1 [ref] | 1 [ref] |
| Intermediate | 1.88 (0.99 to 3.58) | 1.29 (0.53 to 3.17) | 1.42 (0.66 to 3.07) | Moderate | 0.56 (0.31 to 1.02) | 0.32 (0.14 to 0.75) | 0.92 (0.38 to 2.23) |
| High | 2.66 (1.33 to 5.32) | 1.30 (0.45 to 3.76) | 2.77 (1.24 to 6.21) | Favourable | 0.46 (0.25 to 0.83) | 0.29 (0.12 to 0.67) | 0.71 (0.29 to 1.73) |
| P for trend | <0.001 | <0.001 | <0.001 | P for trend | 0.004 | 0.004 | 0.360 |

*CI* confidence interval, *AF* atrial fibrillation, *CAD* coronary artery disease, *VTE* venous thromboembolism, *ISS* ischaemic stroke, *PRS* polygenic risk score, *HLS* healthy lifestyle score, *Ref* reference.
[a]Per one sd increase.
[b]Per one point increase; the Cox model adjusted for age, sex, education level, index of multiple deprivations, ethnicity, genotyping batch, and the first ten genetic principal components. No statistical correction is made for multiple comparisons across subgroups.

## Sensitivity analyses

The results of the sensitivity analysis were consistent with the primary analysis. Of note, the magnitude of genetic association was amplified for all outcomes by using the enhanced PRS (HR: AF 1.58 [1.40 to 1.77]; CAD 1.68 [1.45 to 1.95]; VTE 1.38 [1.09 to 1.76] and ISS 1.40 [0.91 to 2.15]). No noticeable change was found for the sensitivity analyses based on either incident or those hospital admission-related diagnosis in terms of both genetic an lifestyle exposures. The negative control outcome analysis found that the post-COVID-19 risk of diabetes was not associated with an increase of any of four CVE PRSs. (Supplementary Table 5)

## Discussion

This large population-based cohort study found that a higher genetic risk based on the PRS was linearly associated with an increased incidence of acute post-COVID-19 CVE complications. COVID-19 patients with the top 20% of PRSs had a 3.4-fold, 3.3-fold, 2.1-fold and 2.6-fold excess risk of AF, CAD, VTE and ISS, respectively, compared with those with the lowest 20% of PRSs. The identified genetic predisposition was persistent in subgroups of participants who were already at very high risk of CVE and were receiving antithrombotic therapy before infection and in those with a breakthrough infection after full vaccination (2 doses). More importantly, the consistent associations were found during the period of circulating Omicron variants.

We also demonstrated that a composite favourable vs unfavourable healthy lifestyle was associated with 34%, 43%, 35% and 54% lower risk of AF, CAD, VTE and ISS complication after the infection. Furthermore, we observed an additive association between genetic predisposition and lifestyle determinants with regard to post-COVID-19 CVE outcomes. This association was particularly pronounced for AF and CAD, while it was less substantial for the thrombotic events of VTE and ISS.

Our study investigates genetic determinants of COVID-19-related CVE. It differs from previous studies of PRS for adult-onset cardiovascular diseases[15–18] as we targeted people with COVID-19 rather than the general healthy population and we evaluated the risk of acute CVE triggered by SARS-CoV-2 rather than the chronic (≥10 years) or lifetime disease risk. The associations (HR per one SD increase) found in

## Atrial fibrillation

| Characteristics | Subgroup | No. of events | Incidence rate | | Hazard ratio | P interaction |
|---|---|---|---|---|---|---|
| Age | >= 65 years | 1291 | 85.25 | | 1.52 | 1.00 |
| | < 65 years | 106 | 10.84 | | 1.52 | |
| Sex | Male | 926 | 82.80 | | 1.48 | 0.20 |
| | Female | 471 | 34.28 | | 1.59 | |
| BMI | >= 30 | 598 | 100.34 | | 1.82 | <0.01 |
| | < 30 | 799 | 42.13 | | 1.40 | |
| Ethnicity | White | 93 | 35.14 | | 1.73 | 0.15 |
| | Other ethnic | 1304 | 58.54 | | 1.50 | |
| Antithrombotics | User | 748 | 241.01 | | 1.92 | <0.01 |
| | Non–user | 649 | 29.74 | | 1.36 | |
| PCR test settings | Inpatient | 500 | 270.55 | | 2.29 | <0.01 |
| | Outpatient | 897 | 38.87 | | 1.41 | |
| Vaccination | Not or partial | 420 | 78.45 | | 1.80 | <0.01 |
| | Complete | 977 | 49.92 | | 1.45 | |
| | Overall | | | | **1.52** | |

## Coronary artery disease

| Characteristics | Subgroup | No. of events | Incidence rate | | Hazard ratio | P interaction |
|---|---|---|---|---|---|---|
| Age | >= 65 years | 621 | 40.76 | | 1.52 | 0.03 |
| | < 65 years | 112 | 11.45 | | 1.87 | |
| Sex | Male | 529 | 47.08 | | 1.67 | 0.01 |
| | Female | 204 | 14.81 | | 1.33 | |
| BMI | >= 30 | 276 | 45.98 | | 1.69 | 0.17 |
| | < 30 | 457 | 24.04 | | 1.53 | |
| Ethnicity | White | 75 | 28.33 | | 1.46 | 0.46 |
| | Other ethnic | 658 | 29.42 | | 1.59 | |
| Antithrombotics | User | 537 | 171.47 | | 2.18 | <0.01 |
| | Non–user | 196 | 8.96 | | 1.29 | |
| PCR test settings | Inpatient | 273 | 145.21 | | 2.43 | <0.01 |
| | Outpatient | 460 | 19.88 | | 1.44 | |
| Vaccination | Not or partial | 242 | 44.99 | | 1.98 | <0.01 |
| | Complete | 491 | 25.01 | | 1.46 | |
| | Overall | | | | **1.57** | |

## Venous thromboembolism

| Characteristics | Subgroup | No. of events | Incidence rate | | Hazard ratio | P interaction |
|---|---|---|---|---|---|---|
| Age | >= 65 years | 190 | 12.43 | | 1.32 | 0.76 |
| | < 65 years | 54 | 5.52 | | 1.37 | |
| Sex | Male | 145 | 12.85 | | 1.28 | 0.49 |
| | Female | 99 | 7.18 | | 1.40 | |
| BMI | >= 30 | 93 | 15.43 | | 1.47 | 0.27 |
| | < 30 | 151 | 7.93 | | 1.28 | |
| Ethnicity | White | 33 | 12.43 | | 1.35 | 0.94 |
| | Other ethnic | 211 | 9.41 | | 1.33 | |
| Antithrombotics | User | 58 | 18.13 | | 1.39 | 0.69 |
| | Non–user | 186 | 8.50 | | 1.32 | |
| PCR test settings | Inpatient | 74 | 38.80 | | 2.07 | <0.01 |
| | Outpatient | 170 | 7.34 | | 1.23 | |
| Vaccination | Not or partial | 135 | 25.04 | | 1.65 | 0.03 |
| | Complete | 109 | 5.54 | | 1.24 | |
| | Overall | | | | **1.33** | |

## Ischaemic stroke

| Characteristics | Subgroup | No. of events | Incidence rate | | Hazard ratio | P interaction |
|---|---|---|---|---|---|---|
| Age | >= 65 years | 93 | 6.08 | | 1.25 | 0.56 |
| | < 65 years | 11 | 1.12 | | 1.48 | |
| Sex | Male | 64 | 5.66 | | 1.21 | 0.55 |
| | Female | 40 | 2.90 | | 1.37 | |
| BMI | >= 30 | 31 | 5.14 | | 1.50 | 0.26 |
| | < 30 | 73 | 3.83 | | 1.19 | |
| Ethnicity | White | 9 | 3.39 | | 1.44 | 0.69 |
| | Other ethnic | 95 | 4.23 | | 1.26 | |
| Antithrombotics | User | 35 | 10.93 | | 1.32 | 0.81 |
| | Non–user | 69 | 3.15 | | 1.25 | |
| PCR test settings | Inpatient | 34 | 17.76 | | 1.59 | 0.33 |
| | Outpatient | 70 | 3.02 | | 1.23 | |
| Vaccination | Not or partial | 30 | 5.55 | | 1.03 | 0.30 |
| | Complete | 74 | 3.76 | | 1.34 | |
| | Overall | | | | **1.27** | |

**Fig. 2 | Genetic risk in key clinical subgroups.** The Cox model included covariates of age, sex, education level, index of multiple deprivations, ethnicity, genotyping batch, and the first ten genetic principal components, in addition to a polygenic risk score and a multiplicative interaction term of the polygenic risk score with the stratification variable of interest. Each error bar is presented as lower and upper 95% confidence interval and no statistical correction is made for multiple comparisons across subgroups.

previous studies were 2.33 for AF[16], 1.86 for CAD[16], 1.26 for ISS[19], and 1.27 for VTE[18], compared with 1.52 for AF, 1.57 for CAD, 1.27 for ISS, and 1.33 for VTE found here. Our observed associations demonstrate that host genetic variations are an important contributor to CVE development following the infection and highlight underlying genetic interconnections between chronic and post-COVID-19 cardiovascular complications. However, the magnitude of gene association was reduced for some CVE subtypes, suggesting that distinctive pathogenic mechanisms may be involved[2,20], such as the virus directly mediating heart injury by entering cardiomyocytes[21,22]. Importantly, the polygenic variations for VTE were largely retained for predicting COVID-19-related VTE, which echo our previous findings that the monogenic variation, such as factor V Leiden mutation, also predisposed post-COVID-19 VTE complications[6].

The role of genetics in COVID-19 multisystem representations is not yet well-understood. Although many large genome-wide association studies were performed for COVID-19, most focused on SARS-COV-2-induced critical respiratory disorders or diseases severity[23,24]. We leveraged well-developed PRSs and have shown that human polygenic variations affect CVE manifestations after COVID-19. There is also a lack of evidence for the potential beneficial effects of healthy lifestyles on reducing the cardiovascular disease burden in COVID-19. Although many studies have reported that lifestyle factors affected chronic cardiovascular conditions independently of individuals' genetic background before the pandemic[15,17], little is known whether this remains the case in COVID-19 cardiovascular complications. Some studies have observed lower risks of severe COVID-19 among people adopting more favourable behaviours[25,26]. However, these studies concentrated on general health utilisation outcomes such as hospital or ICU admissions, therefore limiting the specificity of the observed associations for CVE. Beyond the clinically relevant outcomes, one recent study of 1981 women found that pre-infection healthy lifestyle was also associated with a substantially lower risk of self-reported post-COVID-19 conditions, known as long COVID[27].

Our findings have implications for clinical responses and public health preparedness against the ongoing COVID-19 pandemic. At the individual level, compared with known general risk factors such as demographic characteristics (e.g., age and sex) and clinical risk factors (e.g., obesity and hypertension), genetic factors might inform more tailored treatment choices to prevent specific COVID-19 complications. For instance, disease-specific PRSs could help doctors identify patients with high genetic risk for arterial thrombosis who would benefit from platelet inhibitors or identify those with high genetic risk for venous thrombosis such as VTE and prioritise them for coagulation cascade suppression therapy[3,28]. Such specific PRSs cannot be achieved using traditional clinical factors alone, such as age, as they were associated with both high VTE risk and high adverse events related to pharmaceutical therapy such as antithrombotics.

Over the past 10 to 15 years, global interest, efforts, and controversies have surrounded PRSs' clinical utility for the primary prevention of non-communicable cardiovascular diseases, such as predicting 10-year risk in the general population[9,11,29]. The potential of a PRS could be magnified in patients with COVID-19 as they have a substantially increased CVE risk, particularly during the initial illness. If a PRS was calculated for everyone at birth and held as part of their health records[30], it could have been used as easily as demographic determinants like age and sex to refine existing approaches to defining subgroups who are particularly vulnerable to COVID-19, possibly providing more timely, personalised shielding advice. Even a small or modest improvement in stratification accuracy might lead to a sizeable population changing their COVID-19 vulnerability category.

Although the genetic risk for post-COVID-19 CVE is inherited, our study showed that acquired healthy behaviours could offset this risk. The US Preventive Services Task Force updated its guidelines in 2022,

## Atrial fibrillation

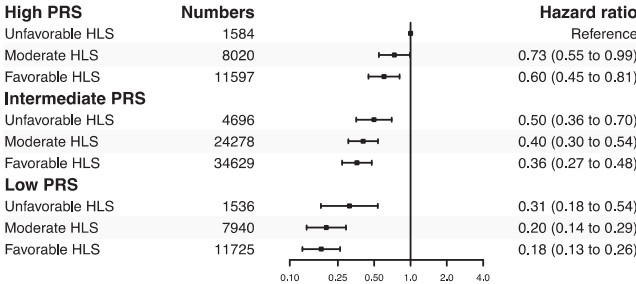

| High PRS | Numbers | | Hazard ratio |
|---|---|---|---|
| Unfavorable HLS | 1584 | | Reference |
| Moderate HLS | 8020 | | 0.73 (0.55 to 0.99) |
| Favorable HLS | 11597 | | 0.60 (0.45 to 0.81) |
| **Intermediate PRS** | | | |
| Unfavorable HLS | 4696 | | 0.50 (0.36 to 0.70) |
| Moderate HLS | 24278 | | 0.40 (0.30 to 0.54) |
| Favorable HLS | 34629 | | 0.36 (0.27 to 0.48) |
| **Low PRS** | | | |
| Unfavorable HLS | 1536 | | 0.31 (0.18 to 0.54) |
| Moderate HLS | 7940 | | 0.20 (0.14 to 0.29) |
| Favorable HLS | 11725 | | 0.18 (0.13 to 0.26) |

## Coronary artery disease

| High PRS | Numbers | | Hazard ratio |
|---|---|---|---|
| Unfavorable HLS | 1558 | | Reference |
| Moderate HLS | 8101 | | 0.92 (0.62 to 1.36) |
| Favorable HLS | 11542 | | 0.64 (0.43 to 0.96) |
| **Intermediate PRS** | | | |
| Unfavorable HLS | 4760 | | 0.62 (0.40 to 0.96) |
| Moderate HLS | 24101 | | 0.39 (0.27 to 0.57) |
| Favorable HLS | 34742 | | 0.34 (0.23 to 0.50) |
| **Low PRS** | | | |
| Unfavorable HLS | 1498 | | 0.38 (0.19 to 0.75) |
| Moderate HLS | 8036 | | 0.24 (0.15 to 0.40) |
| Favorable HLS | 11667 | | 0.20 (0.13 to 0.33) |

## Venous thromboembolism

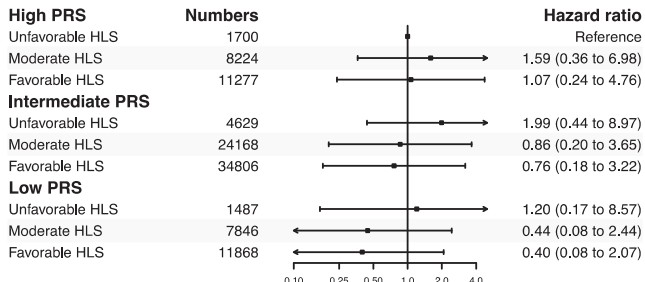

| High PRS | Numbers | | Hazard ratio |
|---|---|---|---|
| Unfavorable HLS | 1569 | | Reference |
| Moderate HLS | 8069 | | 1.61 (0.63 to 4.10) |
| Favorable HLS | 11563 | | 1.02 (0.40 to 2.65) |
| **Intermediate PRS** | | | |
| Unfavorable HLS | 4770 | | 1.07 (0.39 to 2.92) |
| Moderate HLS | 24120 | | 0.82 (0.33 to 2.05) |
| Favorable HLS | 34713 | | 0.74 (0.30 to 1.84) |
| **Low PRS** | | | |
| Unfavorable HLS | 1477 | | 1.30 (0.40 to 4.25) |
| Moderate HLS | 8049 | | 0.70 (0.26 to 1.92) |
| Favorable HLS | 11675 | | 0.40 (0.14 to 1.15) |

## Ischaemic stroke

| High PRS | Numbers | | Hazard ratio |
|---|---|---|---|
| Unfavorable HLS | 1700 | | Reference |
| Moderate HLS | 8224 | | 1.59 (0.36 to 6.98) |
| Favorable HLS | 11277 | | 1.07 (0.24 to 4.76) |
| **Intermediate PRS** | | | |
| Unfavorable HLS | 4629 | | 1.99 (0.44 to 8.97) |
| Moderate HLS | 24168 | | 0.86 (0.20 to 3.65) |
| Favorable HLS | 34806 | | 0.76 (0.18 to 3.22) |
| **Low PRS** | | | |
| Unfavorable HLS | 1487 | | 1.20 (0.17 to 8.57) |
| Moderate HLS | 7846 | | 0.44 (0.08 to 2.44) |
| Favorable HLS | 11868 | | 0.40 (0.08 to 2.07) |

**Fig. 3 | Joint associations between genetic and lifestyle factors.** The Cox model adjusted for age, sex, education level, index of multiple deprivations, ethnicity, genotyping batch, and the first ten genetic principal components and index variable of the nine genetic and lifestyle subgroups. Each error bar is presented as lower and upper 95% confidence interval and no statistical correction is made for multiple comparisons across subgroups.

recommending behavioural counselling for cardiovascular disease prevention for all adults aged 18 years and older[12,14]. Our data support this recommendation by showing that a healthier population lifestyle background can also alleviate the immediate CVE burden after COVID-19, regardless of genetic risk. Notably, a healthy lifestyle contributes to

maintaining better blood coagulability and haemostasis, reducing oxidative damage, increasing blood flow and being responsible for anti-inflammatory effects, all these mechanisms could underlie the lower risk of cardiovascular complications of COVID-19[31]. Nevertheless, this beneficial effect likely takes years to attain for individuals and our findings should not be interpreted as changing behaviour around the time of acute infection.

Our study benefitted from a large population-based cohort, standardised genotyping, quality-controlled genetic data, powered and validated PRS estimates, well-defined measurements of a range of lifestyle factors, PCR-confirmed COVID-19 infection, and reliable and complete linkages to cardiovascular disease outcomes, which together enable these novel findings.

However, several study limitations should be considered. Our PRS was initially built to quantify polygenic risk for any adult-onset cardiovascular disease. It may not reflect the maximum possible genetic contribution to COVID-19 cardiovascular complications, especially given the likelihood of distinct pathological mechanisms involving virus-induced cardiovascular events. Future GWAS studies explicitly designed for COVID-19-related CVE could inform the development of a bespoke PRS and improve predictive performance.

Observational studies that use routinely collected data to ascertain disease outcomes may record overdiagnoses for COVID-19 patients. The ICD records of some clinical events, such as hypertension or diabetes, immediately after COVID-19 infection could be duplicate records of historical conditions instead of a new or activated disease status. However, all of the cardiovascular disease subtypes except for CAD used as outcomes in this study appeared to be temporary and potentially life-threatening. They are unlikely to be coded for without justification in actual clinical practice. Our sensitivity analyses using only incident or hospital-admission-specific CVE also produced findings consistent with the main analysis, precluding this concern.

Demonstrating statistical significance does not guarantee that the PRS is able to provide additional predictive information up on the existing clinical factors only based cardiovascular models, as previous studies have frequently found little agreement between statistical association and predictive performance[32,33]. More modelling research is urgently needed to fill this evidence gap in the contexts of COVID-19.

We used lifestyle behaviour data collected 10 years ago as a surrogate for current lifestyle habits at the time of infection, which is likely subject to misclassification and may have biased any genuine associations toward the null. Reassuringly, all participants at the time of recruitment were middle-aged or older adults whose lifestyle habits should have been well established, suggesting that their habits are likely to have remained consistent over years between recruitment and infection.

Participants in UK Biobank represent a generally healthier population than the general population of the UK and are mostly of European ancestry, which may limit our findings' generalisability beyond this population.

Individuals' genetic predisposition, in the form of a PRS, was associated with short-term risks of CAD, AF, VTE and ISS complications after COVID-19. However, these post-COVID-19 complications were substantially lower among those previously adhering to a healthy lifestyle, independent of their genetic risk. Overall, our findings demonstrate the role of host genetics in determining COVID-19 triggered cardiovascular events, and suggest that intensifying healthy lifestyle interventions in population may help alleviate the elevated cardiovascular burden.

## Methods

### Data sources and COVID-19 population

UK Biobank is a large-scale, population-based prospective cohort of over 500,000 individuals aged 40 to 69 years at recruitment between

2006 and 2010 from across the United Kingdom[34]. Detailed lifestyle information was collected through questionnaires at 22 assessment centres. Affymetrix performed genotype calling based on two closely related purpose-designed arrays (UK BiLEVE Axiom and UK Biobank Axiom) for all participants[35]. Follow-up diseases outcomes were identified through linkage to various electronic health records, covering national primary and secondary care, disease and mortality registries[36]. The reliability, accuracy and completeness of capturing medical conditions using this linkage approach have been validated in previous studies[37,38]. To enable urgent research into COVID-19, additional data from Public Health England's Second Generation Surveillance System has recently been linked to all UK Biobank participants with a bespoke algorithm to ascertain SARS-CoV-2 infection cases[39]. This information includes dates of sample taken and healthcare settings of the polymerase chain reaction (PCR) testing.

In this study, we enroled a cohort of participants from England who survived in March 1, 2020 and had a positive PCR testing result between March 1, 2020, and September 30, 2022. Any new infections happened after the December 1, 2021 was defined as Omicron variants. Participants with missing information on study exposures and covariates of interest at baseline were excluded. All participants in this study provided written informed consent at the UKBB cohort recruitment. This study received ethical approval from the UKBB Ethics Advisory Committee (EAC) under application 65397.

## Polygenic risk score
In May 2022, UK Biobank released a list of polygenic risk scores for 28 diseases, proving their predictive ability to outperform currently published PRS[40]. Two types of PRS, standard and enhanced, were separately developed and validated. To avoid the risk of overfitting, the standard score was developed based only on non-UK-Biobank populations and calculated for all individuals in UK Biobank. In contrast, the model for enhanced score was developed in a proportion of UK Biobank participants, and then computed for those remaining[41]. In this study, we used the standard PRS of coronary artery disease (CAD), atrial fibrillation (AF), venous thromboembolic disease (VTE), and ischaemic stroke (ISS) in the primary analysis and the enhanced PRS in the sensitivity analyses. The continuous PRS was also dichotomised into three risk categories, including the high genetic risk (5th quintile), intermediate genetic risk (2ed-4th quintiles), and low genetic risk (1st quintile) to enable intuitive interpretation and align with previous studies[17,42].

## Healthy lifestyle score
We generated a composite healthy lifestyle score (HLS) by combining nine lifestyle components[43], which consists of smoking status, alcohol drinking, physical activity, television viewing time, sleep duration, intake of fruit and vegetable, intake of oily fish, intake of red meat, and intake of processed meat. Each lifestyle habit was assigned 1 point if considered healthy and 0 point if considered unhealthy. For example, participants with "alcohol drinking daily or almost daily" scored 0, whereas participants with "alcohol drinking ≤ 4 times week" scored 1 for the alcohol drinking element. We summed a total score of all nine lifestyle factors and manually classified people into three lifestyle categories according to their HLS: unfavourable (0–4), moderate (5–6) and favourable (7–9). More details on defining each lifestyle habit are provided in Supplementary Methods.

## Post-COVID-19 CVE
Among participants with COVID-19, we defined the first infection as the index date and followed up for 90 days. We studied four major CVEs (AF, CAD, ISS, and VTE) that were frequently reported as COVID-19-related cardiovascular complications, through linkage to hospital admissions data. International Classification of Diseases 10th Revision (ICD-10) codes were used to capture clinical outcomes and are presented in Supplementary Methods. These ICD-10 codes were the same as those initially used for PRS development to minimise the impact of variation in disease phenotypes between our and previous studies[41]. Data in this study were censored on September 30, 2022.

## Statistical analyses
We used the Cox proportional hazard (PH) model to assess associations between each PRS for CVE and its corresponding post-COVID-19 complication. The PH assumptions were checked based on Schoenfeld residuals and were satisfied. The hazard ratio (HR) and 95% confidence interval (CI) for the continuous PRS (per 1 standard deviation [SD] increase) were estimated by adjusting for age, sex, education level (mapped to the international standard for classification of education, see Supplementary Methods), index of multiple deprivations (IMD, a continuous summary deprivation measurement used in England that contains crime, education, employment, health, housing, income, and living environment)[31], ethnicity, genotyping batch, and the first ten principal components of genetic ancestry. To avoid overadjustment, we did not adjust for previous CVE conditions that are likely to be on the mediating pathway (meditators) for the associations of genetics and lifestyle with post-COVID-19 CVE complications).

We calculated the PRS-CVE association among subgroups of particular clinical relevance, including age (≥65 years or <65 years), sex (female or male), body mass index (≥30 or <30), ethnicity (White or other ethnic groups), recent antithrombotic medication (yes or no), setting for a positive PRC test (inpatient or outpatient/community), and SARS-COV-2 infection type (breakthrough infection after two-dose vaccination or non-breakthrough infection with one-dose or no vaccination). Multiplicative interactions between the continuous PRS and the stratification variables were tested, and P-values are reported. Restricted cubic splines were used to examine possible nonlinear associations for the continuous PRS[44]. Categorical genetic risk was analysed separately and survival curves of COVID-19 patients in each subgroup was depicted using the Kaplan–Meier method. We repeated the association analysis for the lifestyle score in all COVID-19 participants and across different genetic subgroups, with the same Cox regression model, adjusting for age, sex, education, index of multiple deprivations, and ethnicity. Finally, a joint effect was modelled between the categorical genetic and composite lifestyle factors for their associations with the post-COVID-19 complications.

Several bespoke sensitivity analyses were performed to test robustness of main findings. First, we used the enhanced PRS (instead of standard PRS in the primary analysis) to examine the associations among a sub-cohort of UK Biobank participants whose enhanced PRS data are available (only for the genetic exposure). Second, we studied incident CVE complications by excluding participants with a respective CVE occurred within 1 year before the COVID-19 infection (for both genetic and lifestyle exposure). Third, we ascertained the outcomes by only using the first three disease diagnoses that are the main causes for hospital admission and likely represent more critical cases (for both genetic and lifestyle exposure). Fourth, we conducted a negative control outcome experiment for the association between PRS and type 2 diabetes (only for the genetic exposure). The negative control experiments were designed to detect any spurious bias related to study design, cohort construction, and modelling approach.

The analyses were performed using R software version 4.1.2. All statistical tests were 2-sided without adjusting for multiple comparisons. A 95% CI that did not contain unity was considered statistically significant.

## Reporting summary
Further information on research design is available in the Nature Portfolio Reporting Summary linked to this article.

## Data availability

Bonafide researchers can apply for access to individual-level source data from the UK Biobank at http://ukbiobank.ac.uk/register-apply/. The aggregated data supporting the findings of this study are available within the paper and its supplementary information files. The datasets generated during the current study are not publicly available but can be obtained from the corresponding author, provided that the request aligns with the ethical guidelines and privacy regulations.

## Code availability

The code used for this study has been deposited in a public git repository (https://github.com/xjq8065524/Genetics_lifestyle_COVID_outcomes).

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

## Acknowledgements

J.Q.X. is funded through Jardine-Oxford Graduate Scholarship and a titular Oxford Clarendon Fund Scholarship. The research was partially supported by the Oxford National Institute for Health and Care Research (NIHR) Biomedical Research Centre. DPA is funded through a NIHR Senior Research Fellowship (Grant number SRF-2018-11-ST2-004). The views expressed in this publication are those of the author(s) and not necessarily those of the NHS, the NIHR or the Department of Health.

## Author contributions

J.Q.X., Y.L.F. and D.P.A. contributed to the study's design. J.Q.X., D.N., B.Z., Q.F. and C.X.L. were involved in data acquisition, analysis and/or interpretation. J.Q.X., Y.L.F. and A.P.U. drafted the article and made significant contributions. N.J.W., R.P., and D.P.A. provided technical or supervisory support for the project.

## Competing interests

DPA's department has received grant/s from Amgen, Chiesi-Taylor, Lilly, Janssen, Novartis, and UCB Biopharma. His research group has received consultancy fees from Astra Zeneca and UCB Biopharma. Amgen, Astellas, Janssen, Synapse Management Partners and UCB Biopharma have funded or supported training programmes organised by DPA's department. Roger Paredes has participated in advisory boards for Gilead, MSD, ViiV Healthcare, Theratechnologies and Lilly. His institution has received research support from Gilead, MSD, and ViiV Healthcare. The remaining authors declare no competing interests.
