## [Peer Review File · Nature Communications]

Genetic risk, adherence to healthy lifestyle and acute cardiovascular and thromboembolic complications after pre- and Omicron SARS-COV-2 infectionREVIEWER COMMENTS

Reviewer #1 (Remarks to the Author):

The paper from Xie et al. aimed to study the associations between both lifestyle and heritable risk of CV events (CVE) and CV complications after SARS-CoV-2 infection. The authors have found associations between both lifestyle and genetics and post-COVID-19 atrial fibrillation (AF) and coronary artery disease (CAD). Associations were also observed between genetics and venous thromboembolism (VTE) and between lifestyle and ischemic stroke (ISS).

Overall, the paper is interesting and easy to read. There are only very few unclear statements. My comments mainly concern methodological choices and interpretation of the results.

1. The delay of 90 days post-COVID-19 would need to be justified. Is there any evident reason behind this choice? It may be helpful for the reader to provide the distribution of the delay between infection and post-COVID-19 CVE.
2. When studying associations between a factor X and an outcome related here to COVID-19, conditioning on COVID-19 infection (and so on testing) could be prone to collider bias if these factors influence the probability to be tested. One corresponding DAG could be $X \rightarrow \text{COVID test} + \leftarrow \text{severe COVID} \rightarrow \text{CVE}$? Is this causal hypothesis a possibility here and could this lead to bias the association? I first thought this was related to what is described, line 291, as one sensitivity analysis ("we used all participants in the full UK Biobank population..."), but the results were not presented and I am not sure to understand the outcome in this situation. This may need clarification.
3. As soon as in the second paragraph of introduction, we expect a predictive model "to identify individuals at risk" and the authors discussed (in limitations, line 248) that statistical significance does not guarantee predictive performance. Related to that, results of associations though already informative might be insufficiently strong to be predictive. As well, in the supplementary methods, a figure shows predictive performance of PRS using odds-ratio which, again, is rather a measure of association. This warrants further explanation why predictive performance (rather than associations) has not been presented here.
4. This is also in line with the clinical meaning of the results: is it true if we say that about 20 events (2.5% of 422 AF, 3% of 235 CAD and 6% of 29 ISS) would be avoided among the 25000 participants with an infection, if all unhealthy lifestyle participants switched to a healthy lifestyle?
5. The table 3 reports results of the interaction study. There are p-for-interaction that are close to significance and some differences in the association by genetic risk and of genetic risk by healthy / unhealthy lifestyle (Figure 3). Not sure if one can be confident about the absence of interaction. Furthermore, given the linear associations with genetic risk (sFigure 1), interaction tests would gain to be studied using continuous PRSs (seems not the case in table 3 and information seems not to be provided in the methods section).
6. It is well understood that lifestyle score is a composite measure of 9 factors and its use avoid multiple tests. Nevertheless, as an association was observed and described, if the goal is to take measures to prevent CV complications, this would be useful as supplementary material to provide individual associations with lifestyle factors at least when a significant association was observed with the score. This may add knowledge to already existing recommendations, to suggest the main means of action.
7. About the lifestyle score, how the cut-off of >4 was chosen? Should the score not be based on modelling of cardiovascular risk as the PRSs?
8. Authors have explained in Statistical Analyses, on line 334 that "to avoid overadjustment (they did not adjust for previous CVE conditions". This is endorsed. But, this would have been useful to know what does PRS adds to these conditions.

More detailed comments:

1. For all descriptive tables with ethnicity, I suspect a mistake where the rows would have been switched for participants with COVID-19 between white and other ethnic groups.
2. In the results section, on line 87, I would say "more were women" rather than "most" given that the percentage is relatively close to 50%?

Reviewer #2 (Remarks to the Author):

The manuscript uses data from UK Biobank to assess the associations of polygenic risk scores (PRSs), a score of healthy lifestyle and their interaction with the risk of cardiovascular and thromboembolic events (CVE) following COVID-19. The CVE assessed were atrial fibrillation (AF), coronary artery disease (CAD), ischemic stroke (ISS), and venous thromboembolism (VTE).

The results show that the PRSs were associated with higher risk of AF, CAD and VTE after COVID-19, and a healthy lifestyle was associated with lower risk of AF, CAD and ISS. No evidence of interactions between the PRSs and healthy lifestyle score was observed. The authors conclude that population genetics and lifestyle considerably influence cardiovascular complications following COVID-19.

The conclusions should be toned down. "Our study is the first to investigate genetic determinants for COVID-19-related CVE". "We levered well-developed PRSs and proved that human polygenic variations affect CVE manifestations after COVID-19, beyond the severity of COVID-19 disease". The PRSs for CVE are not specific to COVID-related CVE.

They also conclude that "a composite healthy lifestyle ca also benefit acute CVE outcomes after COVID-19 regardless of genetic risk". The results from Table 3 and Figure 3 do not support this. Furthermore, the authors acknowledge in the introduction that obesity is a predictor of COVID-19, and point out as limitation that the lifestyle factors were assessed at recruitment to UKB (i.e., 10 years or more before the outcome). Implications on this should be further discussed.

The authors describe the healthy lifestyle score. However, it is not clear why/how the chosen cut-off point was chosen. The cited publication uses 3 categories: most healthy (0-2), moderately healthy (3-5) and least healthy (6-9).

Some of the sensitivity analyses performed are not clearly described. In one of the analysis, incident cases only were included. Can the number of cases for each disease be described?

A positive control outcome analysis was performed for the association between PRS and CVE among UK Biobank participants without COVID. Could the authors provide more details on the number of participants and number of cases for each disease?

Additionally, a negative control outcome analysis was performed. In the absence of confounding, it would be expected no association between the PRSs and diabetes. However, a negative association between AF PRS and diabetes was observed (HR 0.82, 95%CI 0.69, 0.97). Can the authors expand on the interpretation of this result? Also, can the author provide more details on this analysis (i.e., participants included, definition of diabetes, follow-up period, number of cases).

Some table and figures are not self-explanatory. In sTable 4, are the results adjusted hazard ratios? What is the figure presented in supplementary material (supplementary methods)?

Minor comments:

In accordance to the statement by the American Statistical Association, avoid using "statistically significant" (Ronald L. Wasserstein, Allen L. Schirm & Nicole A. Lazar (2019) Moving to a World Beyond "p < 0.05", The American Statistician, 73:sup1, 1-19, DOI: 10.1080/00031305.2019.1583913).

The distribution of ethnicity in Table 1, sTable 1, sTable 2 and sTable 3 in participants with COVID is incorrect.

REVIEWER COMMENTS

Reviewer #1 (Remarks to the Author):

The paper from Xie et al. aimed to study the associations between both lifestyle and heritable risk of CV events (CVE) and CV complications after SARS-CoV-2 infection. The authors have found associations between both lifestyle and genetics and post-COVID-19 atrial fibrillation (AF) and coronary artery disease (CAD). Associations were also observed between genetics and venous thromboembolism (VTE) and between lifestyle and ischemic stroke (ISS).

Overall, the paper is interesting and easy to read. There are only very few unclear statements. My comments mainly concern methodological choices and interpretation of the results.

Thanks for your positive comments and suggestions to improve our work.

1. The delay of 90 days post-COVID-19 would need to be justified. Is there any evident reason behind this choice? It may be helpful for the reader to provide the distribution of the delay between infection and post-COVID-19 CVE.

Authors' Response:

The recommended follow-up period for acute COVID-19 outcomes is typically 90 days after confirmed infection. This cut-off is largely based on the WHO guidelines and national recommendations, where any symptoms/complications occurring beyond 90 days of infection fall under the scope of Long of Post-Acute COVID. Please refer to the following links for relevant documents:

- UK National Institute for Health and Care Excellence (NICE). <https://www.nice.org.uk/guidance/ng188/resources/covid19-rapid-guideline-managing-the-longterm-effects-of-covid19-pdf-51035515742>
- WHO Long COVID definition. <https://www.who.int/europe/news-room/fact-sheets/item/post-covid-19-condition#:~:text=Definition,months%20with%20no%20other%20explanation>.

2. When studying associations between a factor X and an outcome related here to COVID-19, conditioning on COVID-19 infection (and so on testing) could be prone to collider bias if these factors influence the probability to be tested. One corresponding DAG could be $X \rightarrow \text{COVID test} + \leftarrow \text{severe COVID} \rightarrow \text{CVE}$? Is this causal hypothesis a possibility here and could this lead to bias the association? I first thought this was related to what is described, line 291, as one sensitivity analysis (“we used all participants in the full UK Biobank population...”), but the results were not presented and I am not sure to understand the outcome in this situation. This may need clarification.

Authors' Response:

Thank you for your comment regarding potential collider bias when conditioning on COVID-19 infection and testing in relation to our study of associations with post-COVID-19 CVE. The potential collider bias by conditioning on COVID-19 patients when studying drug's effect on COVID-19 outcomes has been previously discussed by Mathias J etc. [JAMA. 2022;327(13):1282-1283. doi:10.1001/jama.2022.1820].

In our study, after thorough discussions, we did not identify any plausible colliders that would indicate the presence of such bias in our research involving either lifestyle or genetic exposures.

We appreciate further clarification from the reviewer if there are any remaining concerns on this matter.

3. As soon as in the second paragraph of introduction, we expect a predictive model “to identify individuals at risk” and the authors discussed (in limitations, line 248) that statistical significance does not guarantee predictive performance. Related to that, results of associations though already informative might be insufficiently strong to be predictive. As well, in the supplementary methods, a figure shows predictive performance of PRS using odds-ratio which, again, is rather a measure of association. This warrants further explanation why predictive performance (rather than associations) has not been presented here.

Authors' Response:

We appreciate the reviewer's comment about our writing of statistical significance vs predictive performance in the limitations section. We posit that causal inference and prediction modelling may represent two distinct types of studies in medical research. Our investigation was designed to quantify the associations between genetic and lifestyle factors with post-COVID CVE complications, which primarily falls in the former, where effect sizes such as hazard ratios or odds ratios are typically reported. Thus, the predictive performance metrics, such as ROC or R-squared values, were not calculated and beyond the scope of our study objective.

We acknowledge that we may have used the terms "risk factor" and "predictor" interchangeably throughout the manuscript, which may have caused confusion. We have revised the relevant sections in our manuscript to enhance clarity for our readers. Please refer to (Page 3, line 74-82) for the updated text.

4. This is also in line with the clinical meaning of the results: is it true if we say that about 20 events (2.5% of 422 AF, 3% of 235 CAD and 6% of 29 ISS) would be avoided among the 25000 participants with an infection, if all unhealthy lifestyle participants switched to a healthy lifestyle?

Authors' Response:

Thank you for your comment. We have deleted this statement accordingly (Page 6, line 188-196).

5. The table 3 reports results of the interaction study. There are p-for-interaction that are close to significance and some differences in the association by genetic risk and of genetic risk by healthy / unhealthy lifestyle (Figure 3). Not sure if one can be confident about the absence of interaction. Furthermore, given the linear associations with genetic risk (sFigure 1), interaction tests would gain to be studied using continuous PRSs (seems not the case in table 3 and information seems not be provided in the methods section).

Authors' Response:

Thank you for your valuable feedback. We agree with the reviewer and have modified our conclusion regarding interaction effects between genetic and lifestyle factors (Page 6, line 199-202). Also, interaction tests using continuous PRS, as suggested by the reviewer, has been added in the updated manuscript (Page 10, line 400-401).

6. It is well understood that lifestyle score is a composite measure of 9 factors and its use avoid multiple tests. Nevertheless, as an association was observed and described, if the goal is to take measures to prevent CV complications, this would be useful as supplementary material to provide individual associations with lifestyle factors at least when a significant association was observed with the score. This may add knowledge to already existing recommendations, to suggest the main means of action.

Authors' Response:

Thank you for your valuable comment. We agree that providing additional information about individual associations with lifestyle factors could be useful in developing measures to prevent cardiovascular complications. Therefore, we have included the analysis of associations for each lifestyle component in our manuscript and presented the results in **sFigure 2**, as suggested by the reviewer.

7. About the lifestyle score, how the cut-off of >4 was chosen? Should the score not be based on modelling of cardiovascular risk as the PRSs?

Authors' Response:

Thank you for your comment regarding the cut-off value for the lifestyle score. In our updated study, we defined the composite lifestyle based on the distribution of healthy factors among the COVID-19 patients we studied. Specifically, we divided the composite lifestyle into unfavourable (0-4 healthy factors), intermediate (5-6 healthy factors), and favourable (7-9 healthy factors), which is largely aligned with the previous study as cited in the Methods section of our manuscript. The only difference is that we classified people with **0-4** healthy factors as unfavourable, instead of the **0-3** healthy factors, to allow sufficient sample size in this subgroups for more comprehensive stratification analysis.

8. Authors have explained in Statistical Analyses, on line 334 that “to avoid overadjustment (they) did not adjust for previous CVE conditions”. This is endorsed. But, this would have been useful to know what does PRS adds to these conditions.

Authors' Response:

We appreciate your comment and agree that it would have been useful to clarify the role of the PRS in relation to prior CVE conditions. We did not adjust for previous CVE conditions in order to avoid erroneous estimation of associations, since prior CVE conditions could likely act as an important mediator in the pathway between exposure (genetic or lifestyle factors) and post-COVID-19 cardiovascular complications. Yet, to help address the reviewer's question, we alternatively used a restriction approach in the study design by excluding participants who had a CVE diagnosis within the year before infection (a surrogate for active cardiovascular disease status), and produced consistent findings. Please see sensitivity analysis results Page 27 **sTable 5**.

More detailed comments:

1. For all descriptive tables with ethnicity, I suspect a mistake where the rows would have been switched for participants with COVID-19 between white and other ethnic groups.

Authors' Response:

We thank the reviewer for identifying this issue. We have corrected it in our revised manuscript. (Page 12-13 **Table 1, sTable1-3**)

2. In the results section, on line 87, I would say “more were women” rather than “most” given that the percentage is relatively close to 50%?

Authors' Response:

We agree with the reviewer and have made changes accordingly (Page 4, line 101).

Reviewer #2 (Remarks to the Author):

The manuscript uses data from UK Biobank to assess the associations of polygenic risk scores (PRSs), a score of healthy lifestyle and their interaction with the risk of cardiovascular and thromboembolic events (CVE) following COVID-19. The CVE assessed were atrial fibrillation (AF), coronary artery disease (CAD), ischemic stroke (ISS), and venous thromboembolism (VTE).

The results show that the PRSs were associated with higher risk of AF, CAD and VTE after COVID-19, and a healthy lifestyle was associated with lower risk of AF, CAD and ISS. No evidence of interactions between the PRSs and healthy lifestyle score was observed. The authors conclude that population genetics and lifestyle considerably influence cardiovascular complications following COVID-19.

The conclusions should be toned down. “Our study is the first to investigate genetic determinants for COVID-19-related CVE”. “We levered well-developed PRSs and proved that human polygenic variations affect CVE manifestations after COVID-19, beyond the severity of COVID-19 disease”. The PRSs for CVE are not specific to COVID-related CVE.

Authors’ Response:

We appreciate your feedback on our manuscript and agree with your suggestion to tone down the conclusions. We have revised the statements in line with your comments. Please see Page 6 line 188-190, 204, 224-226 in the updated manuscript.

They also conclude that “a composite healthy lifestyle ca also benefit acute CVE outcomes after COVID-19 regardless of genetic risk”. The results from Table 3 and Figure 3 do not support this. Furthermore, the authors acknowledge in the introduction that obesity is a predictor of COVID-19, and point out as limitation that the lifestyle factors were assessed at recruitment to UKB (i.e., 10 years or more before the outcome). Implications on this should be further discussed.

Authors’ Response:

We agree with the reviewer that alterations in lifestyle patterns (either from unfavourable one to favourable one or vice versa) between the period of baseline enrolment and the onset of the pandemic may result exposure misclassification, likely diminishing the estimated protective influence of a healthful lifestyle, if it exists. We have expanded upon the potential implications of this in our discussion and recognized it as a potential limitation of our research, Page 8, line 301-306.

Furthermore, we have revised our statement "a composite healthy lifestyle may also positively influence acute CVE outcomes following COVID-19, independent of genetic risk" in light of the most recent findings. Please our revised Abstract.

The authors describe the healthy lifestyle score. However, it is not clear why/how the chosen cut-off point was chosen. The cited publication uses 3 categories: most healthy (0-2), moderately healthy (3-5) and least healthy (6-9).

Authors’ Response:

We thank the reviewer’s comment. In our initial version, due to the limited cases of certain post-COVID-19 CVE complications, such as ischemic stroke (with only 29 total cases), we were restricted to the analysis of two relatively broad lifestyle categories (healthy versus unhealthy) to ensure adequate statistical power.

However, the revised manuscript, due to the inclusion of much more COVID-19 patients, allows us to study a more granular categorization of lifestyle (favourable, moderate, unfavourable), as suggested by the reviewer and consistent with the methodology as cited.

Some of the sensitivity analyses performed are not clearly described. In one of the analysis, incident cases only were included. Can the number of cases for each disease be described?

Authors' Response:

We have elaborated our sensitivity analyses in the present manuscript. The number of cases for each study outcomes has been included, as suggested, in Page 27 **sTable 5**.

A positive control outcome analysis was performed for the association between PRS and CVE among UK Biobank participants without COVID. Could the authors provide more details on the number of participants and number of cases for each disease?

Authors' Response:

We thank the reviewer's comment. We have removed the positive control analysis in our update version given that the association between the PRS and CVD outcomes has been previously reported predated the pandemic among UK Biobank participants.

Additionally, a negative control outcome analysis was performed. In the absence of confounding, it would be expected no association between the PRSs and diabetes. However, a negative association between AF PRS and diabetes was observed (HR 0.82, 95%CI 0.69, 0.97). Can the authors expand on the interpretation of this result? Also, can the author provide more details on this analysis (i.e., participants included, definition of diabetes, follow-up period, number of cases).

Authors' Response:

Thank you for your insightful comment. In our original analysis, we identified a negative correlation between AF-PRS and diabetes, potentially attributable to an inflated Type 1 error arising from multiple testing. However, upon incorporating more COVID-19 patient data in our updated manuscript, no associations with diabetes were detected. Detailed results, as per the reviewers' request, are reported in Page 27 **sTable 5** and Page 31.

Some table and figures are not self-explanatory. In sTable 4, are the results adjusted hazard ratios? What is the figure presented in supplementary material (supplementary methods)?

Authors' Response:

We thank the reviewer's comment. Indeed, the findings presented in the initial version of sTable 4 were adjusted hazard ratios. We have modified the labelling in the updated manuscript to ensure clarity (now **sTable 5** in the revised version).

Minor comments:

In accordance to the statement by the American Statistical Association, avoid using "statistically significant" (Ronald L. Wasserstein, Allen L. Schirm & Nicole A. Lazar (2019) Moving to a World Beyond " $p < 0.05$ ", The American Statistician, 73:sup1, 1-19, DOI: 10.1080/00031305.2019.1583913).

Authors' Response:

We thank the reviewer's suggestion. We have avoided using this term as much as possible throughout the manuscript except where it is necessary to convey important findings or results.

The distribution of ethnicity in Table 1, sTable 1, sTable 2 and sTable 3 in participants with

COVID is incorrect.

Authors' Response:

Thank you for pointing out this error. We apologize for any confusion this may have caused. We have made necessary corrections accordingly. (Page 12-13 **Table 1, sTable1-3**)

REVIEWERS' COMMENTS

Reviewer #1 (Remarks to the Author):

The revision process was not facilitated by the absence of a version of the manuscript with tracked changes.

Point 1. I'm not really convinced by the justification of the delay provided by the authors. For acute events, the delay post-infection is often considered shorter. Due to the information that most of the events occurred within 30 days post-infection, we may consider the results as valid. Sensitivity analyses would have been useful, in addition to the distribution of the delay (this could be done for example in sTable 4).

Point 2. I stated that, when studying effect of X on CVE, conditioning on positive test is susceptible to open the following path: X -> test & COVID + <- severe COVID -> CVE (there are probably other possible pathways, I'm just providing here an example). To estimate the unbiased causal association between X and CVE, the path must be closed. My question is then: is such a causal relationship possible? I guess (but I may be wrong) that this may be less likely for PRS, and more an issue for lifestyle factors as we could hypothesize that tobacco smokers, TV viewers or fruit/vegetables consumers (for instance), could have a different probability to be tested in case of symptoms. In this situation, not stratifying for COVID test could be a better analysis. Another example, perhaps closer than in the JAMA paper cited by the authors, is what can happen in test-negative designs (e.g. 10.1093/aje/kww063 and 10.1093/aje/kww064). With no further information on the associations between studied factors and SARS-CoV-2 identification, this point could be stated in the limitations.

Point 3. Objective has substantially changed. There are now more in line with the methods used.

Point 4. Statement has been deleted. But if calculation is true, we might ask ourselves how important the results are for Public Health.

Point 5. Again, changes are hard to follow without tracked changes.

Point 6. Ok.

Point 7. Ok.

Point 8. Ok.

Additional issue. Line 134 and figure S1. The linearity test should be presented differently. Indeed, (except if I'm wrong) the test for linearity do not allow to conclude "the association are significantly linear" : there is in general two steps: 1/ we do not reject that the non-linearity term(s) is/are null 2/ considering linearity, there is a significant association. These tests would gain to be better stated.

Minor comments

1. Please, explain SMD in the tables
2. Line 42: characterised "by"
3. Line 112, 139,7 \diamond 1397
4. Line 314, was \diamond were

REVIEWER COMMENTS

Reviewer #1 (Remarks to the Author):

The revision process was not facilitated by the absence of a version of the manuscript with tracked changes.

We should have provided both a clean and a tracked version to facilitate the last review process, and we apologize for any inconvenience that may have been caused if we inadvertently failed to do so.

Point 1. I'm not really convinced by the justification of the delay provided by the authors. For acute events, the delay post-infection is often considered shorter. Due to the information that most of the events occurred within 30 days post-infection, we may consider the results as valid. Sensitivity analyses would have been useful, in addition to the distribution of the delay (this could be done for example in sTable 4).

The definition of the acute period as 30 or 90 days can be quite subjective. The 90-day period has been commonly used in prior medical research, as we mentioned in our previous correspondence with the reviewer. However, in response to the reviewer's concern, we performed a sensitivity analysis. By limiting the follow-up to 30 days, we found the results to be nearly identical. This similarity can also be discerned from the survival curves shown in Figure 1.

Point 2. I stated that, when studying effect of X on CVE, conditioning on positive test is susceptible to open the following path: $X \rightarrow \text{test} \& \text{COVID} + \leftarrow \text{severe COVID} \rightarrow \text{CVE}$ (there are probably other possible pathways, I'm just providing here an example). To estimate the unbiased causal association between X and CVE, the path must be closed. My question is then: is such a causal relationship possible? I guess (but I may be wrong) that this may be less likely for PRS, and more an issue for lifestyle factors as we could hypothesize that tobacco smokers, TV viewers or fruit/vegetables consumers (for instance), could have a different probability to be tested in case of symptoms. In this situation, not stratifying for COVID test could be a better analysis. Another example, perhaps closer than in the JAMA paper cited by the authors, is what can happen in test-negative designs (e.g. 10.1093/aje/kww063 and 10.1093/aje/kww064). With no further information on the associations between studied factors and SARS-CoV-2 identification, this point could be stated in the limitations.

We appreciate the reviewer comment on the need to address collider bias, if it appears in an observational study. However, we have not found any evidence of this in our study. We believe that the phenomenon the reviewer describes aligns more closely with "detection bias". This form of bias refers to the scenario where individuals with certain lifestyles may be more prone to testing, and as a result, are more likely to receive a diagnosis of COVID-19 and/or a subsequent CVE. In our study, we have managed to rule out this form of bias by conditioning specifically on patients who have been diagnosed with COVID-19.

Point 3. Objective has substantially changed. There are now more in line with the methods used.

Thanks.

Point 4. Statement has been deleted. But if calculation is true, we might ask ourselves how important the results are for Public Health.

The calculation of the population's prevented fraction (PFP) relies heavily on the specific proportions of healthy lifestyle habits within the population. However, since our sample population doesn't entirely reflect the general population in terms of these proportions (as a limitation discussed in the manuscript), we have decided to exclude the PFP calculation from our revised version for the sake of accuracy.

Point 5. Again, changes are hard to follow without tracked changes.

We apologise for any inconvenience caused. We will ensure that a version with tracked changes is provided in all subsequent communications.

Point 6. Ok.
Thanks.

Point 7. Ok.
Thanks.

Point 8. Ok.
Thanks.

Additional issue. Line 134 and figure S1. The linearity test should be presented differently. Indeed, (except if I'm wrong) the test for linearity do not allow to conclude "the association are significantly linear" : there is in general two steps: 1/ we do not reject that the non-linearity term(s) is/are null 2/ considering linearity, there is a significant association. These tests would gain to be better stated.

Thanks. We have clarified this in the legend of sFigure 1 and amended the main texts.

Minor comments

1. Please, explain SMD in the tables

Thanks. The full spell of "standardised mean difference" has been added in the figure legend.

2. Line 42: characterised "by"

Thanks.

3. Line 112, 139,7 à 1397

Thanks.

4. Line 314, was à were

Thanks.